# The Association between Decreased Kidney Function and FIB-4 Index Value, as Indirect Liver Fibrosis Indicator, in Middle-Aged and Older Subjects

**DOI:** 10.3390/ijerph18136980

**Published:** 2021-06-29

**Authors:** Kazuko Kotoku, Ryoma Michishita, Takuro Matsuda, Shotaro Kawakami, Natsumi Morito, Yoshinari Uehara, Yasuki Higaki

**Affiliations:** 1Graduate School of Sports and Health Science, Fukuoka University, 8-19-1 Nanakuma, Johnann-ku, Fukuoka 814-0180, Japan; koutoku@fmt.teikyo-u.ac.jp; 2Faculty of Fukuoka Medical Technology, Teikyo University, 6-22 Misakimachi, Omuta, Fukuoka 836-8505, Japan; 3Laboratory of Exercise Physiology, Faculty of Sports of Sports and Health Science, Fukuoka University, 8-19-1 Nanakuma, Johnan-ku, Fukuoka 814-0180, Japan; rmichishita@fukuoka-u.ac.jp (R.M.); skawakami@fukuoka-u.ac.jp (S.K.); ueharay@fukuoka-u.ac.jp (Y.U.); 4The Fukuoka University Institute for Physical Activity, 8-19-1 Nanakuma, Johnan-ku, Fukuoka 814-0180, Japan; 5Department of Rehabilitation, Fukuoka University Hospital, 7-45-1 Nanakuma, Johnan-ku, Fukuoka 814-0133, Japan; gd050010@yahoo.co.jp; 6Fukuoka University Health Care Center, 7-45-1 Nanakuma, Johnan-ku, Fukuoka 814-0133, Japan; moriton@fukuoka-u.ac.jp; 7Department of Cardiology, Fukuoka University School of Medicine, 7-45-1 Nanakuma, Johnan-ku, Fukuoka 814-0133, Japan

**Keywords:** liver fibrosis, decreased kidney function, CKD, FIB-4 index

## Abstract

Liver fibrosis might be linked to the prevalence of chronic kidney disease (CKD). However, there is little information about the association between liver fibrosis and decreased kidney function in middle-aged and older subjects. We aimed to evaluate the influence of liver fibrosis on the incidence or prevalence of CKD stage 3–5 in a retrospective cross-sectional study (Study 1, *n* = 806) and a 6-year longitudinal study (Study 2, *n* = 380) of middle-aged and older subjects. We evaluated liver fibrosis using the Fibrosis-4 (FIB-4) index and kidney function using the estimated glomerular filtration rate (eGFR) of all subjects. All subjects were divided into four groups on the basis of their FIB-4 score quartiles (low to high). In the Jonckheere–Terpstra trend test of Study 1, the eGFR decreased significantly from the lowest group to the highest group (*p* < 0.001). The Kaplan–Meier survival curve in Study 2 showed that the cumulative prevalence of CKD stage 3–5 was higher in the third quartile than the other quartiles. Our results suggest that liver fibrosis could be a useful indicator for the prevalence of CKD, even within a relatively healthy population, although liver fibrosis was not an independent risk factor.

## 1. Introduction

Chronic liver disease includes the progressive destruction and regeneration of liver parenchyma, which results in fibrosis and cirrhosis that is caused by inflammation and abnormal metabolism [1]. Chronic kidney disease (CKD) has multiple traditional and nontraditional risk factors that are similar to those of chronic liver disease, such as insulin resistance and chronic inflammation [2]. Previous studies have shown that non-alcoholic fatty liver disease (NAFLD) is associated with CKD complication [3,4,5,6], suggesting that liver fibrosis might predict the early stage of CKD development. A previous study showed that NAFLD was independently associated with a high prevalence of microalbuminuria and an increased risk of CKD; if the disease is identified early, it can be appropriately managed using standard medical therapies [6].

The Kidney Disease: Improving Global Outcomes (KDIGO) guidelines indicate that the kidneys are the target organ of many diseases, and significant aggravation or systemic pathophysiological processes can begin through their complex functions and effects on body homeostasis [7]. The causes of CKD have been reported to be complex and include common diseases such as hypertension, metabolic syndrome, diabetes, etc., that mainly affect the kidney [7,8,9]. As mentioned earlier, an increasing amount of evidence suggests that chronic liver disease is tightly linked to metabolic syndrome, which may be associated with many comorbidities such as insulin resistance, cardiovascular disease (CVD), and acute kidney injury [1].

It is known that kidney and liver dysfunction progress through common risk factors such as obesity, type 2 diabetes, hypertension, and dyslipidemia [5]. Although it is known that kidney and liver fibrosis are promoted by these common risk factors, the relationship between the two remains unclear. This study focused on relatively healthy middle-aged and older subjects. We believe that clarifying the relationship between kidney dysfunction and liver fibrosis, even in relatively healthy subjects, can contribute to the onset of CKD and the progression to NAFLD and NASH prevention in the future. A previous study showed that the Fibrosis-4 index (FIB4-index; a simple index for assessing liver fibrosis) was superior to other tested noninvasive markers of fibrosis in Japanese patients with NAFLD, with a high negative predictive value for excluding advanced fibrosis [10].

The aim of this study was to determine the association between the progression of liver fibrosis, which was evaluated using the FIB-4 index and the estimated glomerular filtration rate (eGFR), on the incidence of CKD stage 3–5 in a retrospective cross-sectional study and a 6-year longitudinal study in middle-aged and older subjects.

## 2. Materials and Methods

### 2.1. Setting

This study was based on the previous cohort study, and all data were collected as described for our previous study in 2017 [11].

### 2.2. Study Design

#### 2.2.1. Study 1

A retrospective cross-sectional study identified the baseline clinical characteristics that were associated with CKD stage 3–5 in middle-aged and older subjects.

#### 2.2.2. Study 2

A retrospective longitudinal cohort study identified baseline clinical characteristics that predict the progression of CKD stage 3–5 in middle-aged and older subjects.

### 2.3. Study Population

There were 4919 adults who received a periodic health checkup at a health center at Fukuoka University from 2008 to 2014. All subjects aged 29 to 72 years were included in the study. All subjects provided informed consent to participate after agreeing with the purpose, methods, and significance of the study. The study conformed to the Declaration of Helsinki guidelines and was approved by the Ethics Committee of Fukuoka University (No. 11-08-01). All subjects in these studies were divided into four groups on the basis of their FIB-4 score quartiles, as follows: Study 1—Group A was 0.78 or less (<0.78), Group B was between 0.78 and 1.01 (≥0.78, <1.01), Group C was between 1.01 and 1.29 (≥1.01, <1.29), and Group D was 1.29 or greater (≥1.29); Study 2—Group A was 0.80 or less (<0.80), Group B was between 0.80 and 1.02 (≥0.80, <1.02), Group C was between 1.02 and 1.32 (≥1.02, <1.32), and Group D was 1.32 or greater (≥1.32).

#### 2.3.1. Study 1

A flowchart for participant inclusion in Study 1 is shown in Figure 1. Subjects undergoing dialysis for kidney failure and with a previous history of liver disease were excluded from the analysis. Eight hundred six people were eligible for Study 1.

#### 2.3.2. Study 2

A flowchart for participant inclusion in Study 2 is shown in Figure 2. Subjects with a previous history of liver disease and decreased kidney function (eGFR estimated by the Japanese GFR inference formula < 60 mL/min/1.73 m^2^) were excluded from the analysis. Three hundred eighty subjects with no missing information over the previous 6 years were eligible for Study 2.

### 2.4. Blood Sampling, Blood Pressure, and Anthropometry Measurements

Physical characteristics were examined for each subject. The physical factors included age, sex, systolic blood pressure (SBP), diastolic blood pressure (DBP), and body mass index (BMI). Plasma hemoglobin, platelet count, aspartate aminotransferase (AST), alanine aminotransferase (ALT), γ-glutamyl transferase (γ-GTP), blood urea nitrogen (BUN), uric acid, fasting glucose, hemoglobin A_1_c (HbA_1_c), serum creatinine, triglyceride levels, high-density lipoprotein cholesterol (HDL-C), and low-density lipoprotein cholesterol (LDL-C) levels were measured for all subjects. The AST-to-ALT ratio was calculated as AST/ALT.

### 2.5. Definitions

Kidney function: The CKD grade was classified in accordance with the eGFR. The eGFR level was calculated using the following Japanese GFR inference formula: eGFR (mL/min/1.73 m^2^) = 194 × serum creatinine (mg/dL)^−1.094^ × age (years)^−0.287^ (×0.739, if women) [12]. In this study, CKD stage 3–5 was defined in accordance with the following Japanese Society Nephrology definition: eGFR < 60 mL/min/1.73 m^2^.

Liver function: Pathological examination of a liver biopsy specimen is required to diagnose liver fibrosis. However, because liver biopsy is invasive, various scoring systems were used that are associated with liver fibrosis induction, and these included age, AST-to-ALT ratio, and platelet count. The FIB-4 index has been reported to be the liver fibrosis system with the most valid AUROC (area under the receiver operating characteristic) [3,10,13], and it was assessed for each subject. It was calculated using the following formula: [age × AST (IU/L)/platelet count (×10^9^/L) × ALT (IU/L)] [10,13,14].

### 2.6. Assessment of Lifestyle Behavior

The subjects’ lifestyle behaviors regarding drinking and smoking habits were selected for the present study based on the standardized self-administered questionnaire from the National Health Promotion Program. Each measurement method was followed as described in our previous study [11].

### 2.7. Statistical Analysis

The data were expressed as the mean ± standard deviation (SD). We evaluated differences between the four FIB-4 index groups using a one-way analysis of variance (ANOVA). Gender, drinking and smoking habits, taking antihypertensive drugs, lipid-lowering drugs, or anti-hyperglycemic drugs were evaluated using the chi-square test.

#### 2.7.1. Study 1

Differences in the incidence of decreased kidney function among the four FIB-4 index groups were visualized using a Jonckheere–Terpstra trend test. To explore the risk factors that are associated with CKD stage 3–5, univariable and multivariable logistic regression models were used. Adjusted variables were chosen on the basis of previous findings [15] and the outcome of the one-way ANOVA. Age, BMI, HDL-C, triglycerides, fasting glucose, and FIB-4 index group were reported to be associated with the incidence of decreased kidney function [15].

#### 2.7.2. Study 2

Differences in the incidence of decreased kidney function among the four FIB-4 index groups were visualized using Kaplan–Meier curves and the log-rank test. A Cox proportional hazards regression model was used to predict the incidence of CKD stage 3–5 for middle-aged and older subjects using the parameters as categorical variables. Hazard ratios were initially adjusted for age, BMI, HDL-C, triglycerides, fasting glucose, and FIB-4 index group at baseline. A probability value < 0.05 was considered to indicate statistical significance. All statistical analyses were performed using SPSS software v. 24 (IBM Corp., Armonk, NY, USA).

## 3. Results

### 3.1. Study 1

#### 3.1.1. Study Population Characteristics

The subjects’ clinical and biochemical features are presented in Table 1. Eight hundred six subjects were included in this analysis. The overall mean age of the cohort was 49.9 ± 8.8 years. Among these subjects, 217 (26.9%) were women, 88 (10.9%) were taking antihypertensive drugs, 61 (7.6%) were taking lipid-lowering drugs, and 25 (3.1%) were taking anti-hyperglycemic drugs. The mean eGFR was 76.7 ± 13.2 mL/min/1.73 m^2^, and the mean FIB-4 index score was 1.10 ± 0.48 (Table 1). There were significant differences in gender, age, eGFR, FIB-4 index, AST/ALT ratio, BMI, SBP, DBP, platelet count, HDL-C, BUN, creatinine, fasting glucose, and HbA_1c_ among the four groups.

In the Jonckheere–Terpstra trend test, the eGFR decreased significantly from Group A to Group D.

#### 3.1.2. Association between Liver Fibrosis and the Prevalence of CKD Stage 3–5

A stratified analysis of FIB-4 index groups as an independent variable was conducted (Table 2, Model 1). Group C showed 4.136 times higher and Group D showed 3.775 times higher rates of CKD stage 3–5 than Group A. A stratified analysis using the FIB-4 index, age, gender, BMI, HDL-C, triglycerides, and fasting glucose as independent variables was also conducted (Table 2, Model 2). Age was thought to be an independent variable that affected CKD stage 3–5 onset.

### 3.2. Study 2

#### 3.2.1. Study Population Characteristics Categorized by the FIB-4 Index

Subjects’ clinical and biochemical features are presented in Table 3. Three hundred eighty subjects were included in this analysis. The overall mean age for the cohort was 50.5 ± 7.0 years. Among these subjects, 101 (26.6%) were women, 38 (10.0%) were taking antihypertensive drugs, 8 (2.1%) were taking antihyperglycemic drugs, and 26 (6.8%) were taking lipid-lowering drugs. The mean eGFR was 77.8 ± 10.3 mL/min/1.73 m^2^, and the mean FIB-4 index score was 1.1 ± 0.5. There were significant differences in age, eGFR, FIB-4 index, AST/ALT ratio, BMI, platelet count, HDL-C, LDL-C, and triglycerides among the four groups (Table 3). The AST/ALT ratio was significantly different between Groups A, C, and D. The BMI was significantly different between Groups A and D. Additionally, HDL and LDL cholesterol were significantly different between Group D and Groups A, B, and C. Triglycerides were significantly different between Groups C and D. Moreover, fasting glucose levels in Groups A and C were significantly different.

#### 3.2.2. Clinical Utility of the FIB-4 Index to Predict the Prevalence of CKD Stage 3–5

Figure 3 shows the cumulative incidence and relative risk of the prevalence of CKD stage 3–5 over a six-year follow up period in the study participants. On categorization by FIB-4 index group, the Kaplan–Meier survival curve showed that the cumulative prevalence of CKD stage 3–5 was higher in FIB-4 index Group C than in Groups A, B, and D. The log-rank test result was 0.06.

Table 4 shows the prevalence of CKD stage 3–5. Because only a stratified group of the FIB-4 index was input as an independent factor, the prevalence of CKD stage 3–5 in Group C was 2.237 times higher than in Group A (Table 4, model 1). In Study 2, a stratified group including the FIB-4 index, gender, age, BMI, triglycerides, HDL-C, and fasting glucose as independent factors (Table 4, model 2) was used. In this analysis, age was an independent variable for the prevalence of CKD stage 3–5.

## 4. Discussion

We researched whether the progression of liver fibrosis was a factor for kidney function deterioration in a retrospective cross-sectional study and a 6-year longitudinal study in middle-aged and older subjects. We found that liver fibrosis was not an independent risk factor of the prevalence of CKD stage 3–5 in middle-aged and older subjects. However, the FIB-4 index and eGFR showed a liner relationship in Study 1. Additionally, when the FIB-4 index was greater than 1.01 (Groups C and D), the prevalence of CKD stage 3–5 increased more than it did for groups with a FIB-4 index that was less than 1.01. These results suggested that the FIB-4 index could be a useful indicator to predict for the prevalence of CKD stage 3–5.

### 4.1. Association between Liver Fibrosis and Decreased Kidney Function

Liver fibrosis is the proliferation of connective tissue that occurs in response to chronic, recurring hepatocellular injury, and it is thought to progress when chronic inflammation is present [1]. Because inflammation is also a factor in the development of CKD [2], it is possible that liver fibrosis caused by inflammation also affects the kidneys and causes CKD. The relationship between liver and kidney pathogenesis is also considered to be a clinical issue, e.g., hepatorenal syndrome [16]. Previous studies indicate that as liver fibrosis progresses, it becomes diseases such as NASH or NAFLD [3,4,5,6,17]; although it is difficult to determine if NAFLD directly causes CKD, it is indirectly related to the onset. In the present study, as liver fibrosis progressed, the prevalence of CKD stage 3–5 morbidity increased, and it is possible that such results were related to each other.

The progress of liver fibrosis is considered to be the progress of chronic inflammation in hemodynamics; chronic inflammation also has an adverse effect on the kidneys [2]. Although the complex interactions among NAFLD, inflammation, oxidative stress, impaired glucose tolerance, atherogenic, and dyslipidemia make it difficult to unravel the specific role of the liver and the underlying mechanisms responsible for the association between NAFLD and the risk of developing CKD, it is clear that chronic inflammation caused by insulin resistance, such as obesity, diabetes, and NAFLD or NASH, induces glomerulosclerosis [3,18]. A previous study also demonstrated that a significant, positive association between NAFLD and CKD was prevalent in the univariate analysis; however, this relationship was attenuated by adjusting for features of metabolic syndrome [19]. A previous review also suggested that it was uncertain whether NAFLD was associated with a specific type of kidney disease, although it is reasonable to assume that NAFLD might promote kidney damage, mostly thorough accelerated atherothrombosis [20].

It is clear that patients with NAFLD or NASH exhibit the typical features of metabolic syndrome and have a myriad of other emerging risk factors and risk markers for CKD [2]. Insulin resistance increases free fatty acids and causes systemic vasculitis. The inflammatory cytokines caused by this chronic inflammation not only cause liver fibrosis through the bloodstream, but also act on the kidney (which is a distant tissue), so that the renal blood vessels also cause chronic inflammation. Specifically, chronic inflammation, which causes liver fibrosis, also affects the kidneys at the same time [21,22,23].

The previous study suggested that it was of great importance to implore public health efforts and strategies to curb the epidemics of obesity and metabolic syndrome. Otherwise, the prevalence of NAFLD and CKD will continue to rise [19]. Kidney function in patients with liver fibrosis, such as NAFLD, should be evaluated and followed-up. Additionally, in the present study, when the FIB-4 index exceeded 1.01, the prevalence of CKD stage 3–5 morbidity doubled. Thus, it might be better to follow liver and kidney function using a FIB-4 index of 1.01. In a previous study, discussions on setting the cutoff point for the FIB-4 index suggested that 1.105 should be regarded as a measure of liver fibrosis progression and an initial decrease in kidney function [3]. No previous studies focused on the FIB-4 index and clarified the relationship between decreased kidney function and liver fibrosis. This is the first study of the relationship between them. As long as the FIB-4 index is expected to be useful for evaluating liver fibrosis [10], it could be a useful indicator for evaluating the prevalence of CKD stage 3–5 morbidity. Following-up on liver fibrosis and decreased kidney function progression at the same time would be beneficial. This could be used during specific health checkups and in certain health guidelines because both the FIB-4 index and eGFR are widely used.

### 4.2. Metabolic Disorder and Arteriosclerosis Induced Deterioration of Kidney and Liver Structure and Function

Liver fibrosis and kidney dysfunction are related and develop together. The common causes of liver fibrosis and kidney dysfunction need to be clarified. Previous studies showed that many factors cause liver fibrosis and kidney disfunction, such as glycolipid metabolism [24,25]. Specifically, not only NAFLD but also CKD are strongly associated with obesity, insulin resistance, and type 2 diabetes [2,24,25]. In the present study, a higher fasting glucose induced a higher FIB-4 index and lower eGFR. This suggests that a metabolic disorder, such as metabolic syndrome, would be a progression factor for liver fibrosis and the prevalence of CKD stage 3–5. Additionally, we determined that the highest prevalence of CKD stage 3–5 was in FIB-4 index Group C, which showed the highest fasting glucose and a lower HDL-C. This finding also suggests that the major predictor must be glucose metabolism and lipid metabolism. Our results suggest that the evaluation of liver fibrosis by means of non-invasive markers is especially important in patients with diabetes and metabolic syndrome for preventing further deterioration.

HDL-C is associated with a lower risk of coronary heart disease and arteriosclerosis [26,27]. Moreover, decreased eGFR was independently associated with decreased HDL-C levels [28,29,30]. HDL-C has anti-inflammatory, anti-oxidative, and anti-apoptotic functions and improves endothelial function; these functions are considered to be anti-atherogenic [26,27]. In the present study, a higher HDL-C effect decreased the risk of CKD stage 3–5 in older age groups. These results suggest that HDL-C could be a prevention factor for liver fibrosis and decreased kidney function.

### 4.3. Age-Related Deterioration of Kidney and Liver Structure and Function

Age and age-related changes to kidney and liver structure and function are the main risk factors for major debilitating and life-threatening conditions, including CVD [31,32,33]. Notably, old age is a risk factor for NAFLD, CKD, and type 2 diabetes, and NAFLD and CKD are major factors for CVD [20]. The present study also shows that eGFR decreases and liver fibrosis increases with aging. Kidney and liver function are the main predictors of longevity; thus, preventing deteriorating kidney and liver function might slow liver fibrosis and CKD stage 3–5 prevalence during aging. In the present study, the group with a higher FIB-4 index score had a greater risk for CKD stage 3–5 prevalence than lower FIB-4 index groups. As discussed above, metabolic disorder and arteriosclerosis might affect kidney and liver function. This result suggests that decreased kidney function and liver fibrosis might be more strongly affected by metabolic disorder and arteriosclerosis than aging.

Most importantly, not only are liver fibrosis and decreased kidney function induced by aging, but they are also accelerated by chronic inflammation such as metabolic disorders and insulin resistance. The early detection of these declines in kidney function, via signs of advanced liver fibrosis, may prevent further complications such as CVD. The FIB4-index was positively correlated with cardiovascular risk scores in patients with chronic liver disease [34]. NAFLD, CKD, and type 2 diabetes are all major risk factors for CVD; therefore, the FIB4-index can be a useful non-invasive indicator for the early detection of these diseases.

### 4.4. Study Limitations and Clinical Implications

Our study had several limitations. First, this was an observational study in middle-aged and older subjects in just one cohort. Therefore, the results may not be generalizable to all people. Second, insulin resistance was not assessed, and thus, it is not clear if there is another association between CKD and the FIB-4 index. Third, although the FIB-4 index is useful for diagnosing liver fibrosis, the gold standard is liver biopsy, which was not evaluated. More studies are needed to determine these relationships. Finally, the subjects of this study were relatively healthy middle-aged and older people, and most of the eGFR and Fib4 values were within the normal range.

However, this was the first study to evaluate FIB-4 index and CKD stage 3–5 prevalence in middle-aged and older subjects. In Japan, not only the number of patients with the introduction of dialysis but also cirrhotic NASH associated with liver fibrosis are steadily increasing [35,36]. Especially, survival in NASH is lower than the expected survival of the matched general population, due to the higher prevalence of cardiovascular and liver-related deaths [34]. We consider that capturing a decrease in eGFR and an increase in FIB-4 index in a relatively healthy population, even within the normal range, is important from the perspective of introducing dialysis and cirrhotic NASH prevention in the future. The present study suggests that liver fibrosis and decreased kidney function are associated with aging. However, it is possible for aging people to reduce CKD stage 3–5 prevalence through a healthy lifestyle, such as preventing increasing insulin resistance and increasing HDL-C levels.

## 5. Conclusions

Our retrospective study evaluated the association of liver fibrosis and other risk factors with the prevalence of CKD stage 3–5 during a 6-year follow-up in middle-aged and older subjects. We found that liver fibrosis was not an independent risk factor of the prevalence of CKD stage 3–5 in middle-aged and older subjects. Although it was not significant, higher FIB-4 index groups (FIB-4 index > 1.01) tended to show more CKD stage 3–5. These results suggested that liver fibrosis could be a useful indicator for the prevalence of CKD, even within a relatively healthy population.

## Figures and Tables

**Figure 1 ijerph-18-06980-f001:**
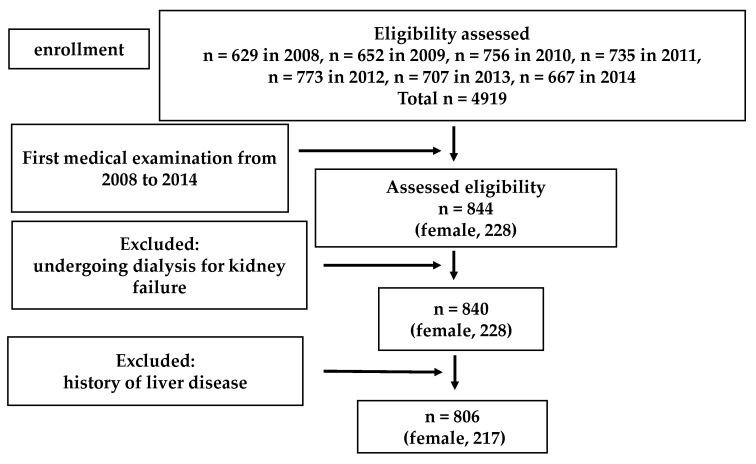
Flow chart of the Study 1 population selection.

**Figure 2 ijerph-18-06980-f002:**
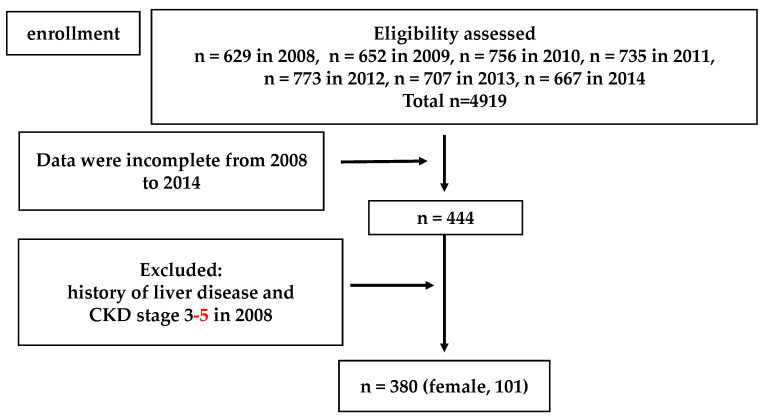
Flow chart of the Study 2 population selection.

**Figure 3 ijerph-18-06980-f003:**
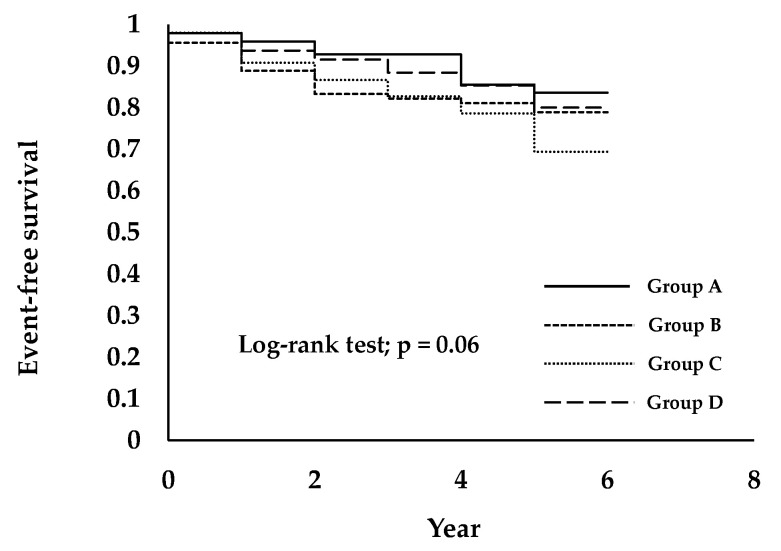
Prevalence of CKD stage 3–5 is associated with a higher FIB-4 index group in healthy people. Subjects were divided into four groups, as follows: Group A was 0.80 or less (<0.80), Group B was between 0.80 and 1.02 (≥0.80, <1.02), Group C was between 1.02 and 1.32 (≥1.02, <1.32), and Group D was 1.32 or greater (≥1.32).

**Table 1 ijerph-18-06980-t001:** The characteristics of subjects in Study 1.

	Total (*n* = 806)			A Group FIB-4 < 0.78 (*n* = 203)			B Group 0.78 ≤ FIB-4 < 1.01 (*n* = 199)			C Group 1.01 ≤ FIB-4 < 1.29 (*n* = 202)			D Group 1.29 ≤ FIB-4 (*n* = 202)			One-Way ANOVA *p* Value	Tukey HSD	Chi-Square *p* Value	Jonckheere–Terpstra Trend Test
Age (y)	49.9	±	8.8	42.7	±	5.9	47.4	±	7.6	52.4	±	7.2	57.3	±	6.8	<0.001 **	Group A < Group B **, C **, D **		
Gender (female%)	26.9			33.5			28.6			24.3			21.3					0.033*	
BMI (kg/m^2^)	22.8	±	3.2	23.0	±	3.2	22.8	±	3.1	23.1	±	3.4	22.3	±	2.9	0.035 *	Group C > Group D *		
SBP (mmHg)	123.3	±	16.0	119.5	±	13.9	121.6	±	16.0	125.2	±	17.3	126.8	±	15.7	<0.001 **	Group A < Group C *, D **		
DBP (mmHg)	78.9	±	11.5	75.9	±	11.7	78.8	±	11.4	79.5	±	11.5	81.2	±	10.7	<0.001 **	Group A < Group C *, D **		
Hemoglobin (g/dL)	14.6	±	1.3	14.5	±	1.4	14.7	±	1.3	14.5	±	1.4	14.7	±	1.4	0.412			
Platelet count (×10^4^/μL)	23.3	±	5.1	27.9	±	5.1	23.9	±	3.6	22.5	±	3.3	18.8	±	3.5	<0.001 **	Group A > Group B, C, D		
AST (U/L)	23.2	±	8.0	20.6	±	6.7	21.6	±	6.4	23.6	±	6.7	27.2	±	10.1	<0.001 **	Group A, B < Group C **, D **		
ALT (U/L)	25.7	±	16.1	26.9	±	17.4	25.9	±	18.3	24.7	±	13.1	25.3	±	15.0	0.551			
AST/ALT ratio	1.06	±	0.38	0.9	±	0.3	1.0	±	0.3	1.1	±	0.4	1.2	±	0.4	<0.001 **	Group A, B < Group C **, D **		
γ-GTP (U/L)	44.6	±	51.9	40.9	±	43.6	39.1	±	38.1	47.2	±	58.6	51.1	±	62.7	0.072			
Total cholesterol (mg/dL)	202.8	±	31.6	201.8	±	31.6	203.1	±	32.7	203.2	±	31.5	203.0	±	30.7	0.968			
Triglyceride (mg/dL)	110.0	±	79.3	117.1	±	98.8	111.0	±	75.2	114.2	±	78.9	97.8	±	57.8	0.072			
HDL-C (mg/dL)	61.4	±	15.3	59.6	±	14.8	59.5	±	14.3	61.8	±	15.9	64.8	±	15.8	0.001 **	Group A, B < Group D *		
LDL-C (mg/dL)	118.1	±	28.1	119.3	±	28.8	121.0	±	28.9	116.8	±	26.9	115.2	±	27.8	0.168			
Fasting glucose(mg/dL)	97.6	±	18.1	93.2	±	10.3	95.2	±	14.0	101.0	±	22.9	101.2	±	20.9	<0.001 **	Group A < Group C **, D **		
HbA1c (%)	5.5	±	0.6	5.4	±	0.4	5.4	±	0.4	5.6	±	0.8	5.6	±	0.8	<0.001 **	Group A, B < Group C *, D *		
BUN (mg/dL)	13.4	±	3.3	12.4	±	3.1	13.1	±	2.9	13.7	±	3.3	14.3	±	3.6	<0.001 **	Group A < Group C, D **		
Uric acid (mg/dL)	5.5	±	1.4	5.5	±	1.5	5.6	±	1.4	5.5	±	1.3	5.6	±	1.4	0.792			
Serum creatinine (mg/dL)	0.8	±	0.2	0.8	±	0.2	0.8	±	0.1	0.9	±	0.4	0.8	±	0.1	0.024 *	Group A < Group C *		
FIB-4 index	1.10	±	0.48	0.64	±	0.09	0.88	±	0.07	1.14	±	0.09	1.74	±	0.47	<0.001 **	Group A < Group B **, C **, D **		
eGFR (mL/min/1.73 m^2^)	76.7	±	13.2	82.3	±	13.0	77.2	±	12.2	73.9	±	13.5	73.3	±	11.9	<0.001 **	Group A > Group B **, C **, D **		*p* < 0.00 **
Smoking habit (yes/no; *n*, %)	117 (14.5)/689 (85.5)			35 (17.2)/168 (82.8)			23 (11.6)/176 (88.4)			31 (15.3)/171 (84.7)			28 (13.9)/174 (86.1)					0.423	
Drinking habit (yes/no; *n*, %)	576 (71.6)/229 (28.4)			140 (69.3)/62 (30.7)			138 (69.3)/61 (30.7)			150 (74.3)/52 (25.7)			148 (73.3)/54 (26.7)					0.574	
Antihypertensive drugs (yes/no; *n*, %)	88 (10.9)/718 (89.1)			11 (5.4)/192 (94.6)			18 (9.0)/181 (91.0)			30 (14.9)/172 (85.1)			29 (14.4)/173 (85.6)					0.005 *	
Lipid-lowering drugs (yes/no; *n*, %)	61 (7.6)/745 (92.4)			10 (4.9)/193(95.1)			11 (5.5)/188 (94.5)			24 (11.9)/178 (88.1)			16 (7.9)/186 (92.1)					0.035 *	
Anti-hyperglycemic drugs (yes/no; *n*, %)	25 (3.1)/781 (96.9)			1 (0.5)/202 (99.5)			3 (1.5)/196 (98.5)			10 (5.0)/192 (95.0)			11 (5.4)/191 (94.6)					0.007 *	

The data are expressed as the mean and standard deviation (SD). We evaluated differences between the four FIB-4 index groups using a one-way analysis of variance. Gender, drinking, and smoking habits; and taking antihypertensive drugs, lipid-lowering drugs, or anti-hyperglycemic drugs were evaluated using a chi-square test. Differences in the incidence of CKD stage 3–5 among the four FIB-4 index groups were visualized using a Jonckheere–Terpstra trend test. A probability value <0.05 was considered to indicate statistical significance. * *p* < 0.05, ** *p* < 0.01. BMI; body mass index, SBP; systolic blood pressure, DBP; diastolic blood pressure, AST; aspartate aminotransferase, ALT; alanine aminotransferase, γ-GTP; γ-glutamyl transferase, HDL-C; high-density lipoprotein cholesterol, LDL-C; low-density lipoprotein cholesterol, HbA1c; hemoglobin A1c, BUN; blood urea nitrogen, eGFR; estimated glomerular filtration rate.

**Table 2 ijerph-18-06980-t002:** Multivariable analysis of odds ratio for CKD stage 3–5 in Study 1.

Model	Characteristics	B (S.E.)	Wald	Odds Ratio	95% CI	*p* Value
1	FIB-4 index group B	0.393 (0.503)	0.610	1.481	0.553–3.972	0.435
	FIB-4 index group C	1.420 (0.438)	10.494	4.136	1.752–9.766	0.001 **
	FIB-4 index group D	1.328 (0.442)	9.039	3.775	1.588–8.976	0.003 **
2	FIB-4 index group B	−0.066 (0.529)	0.016	0.936	0.332–2.641	0.901
	FIB-4 index group C	0.622 (0.495)	1.576	1.863	0.705–4.919	0.209
	FIB-4 index group D	0.322 (0.550)	0.342	1.380	0.469–4.057	0.559
	gender	−0.131 (0.432)	0.092	0.877	0.376–2.045	0.761
	age	0.084 (0.022)	14.724	1.088	1.042–1.135	<0.001 **
	BMI	0.091 (0.047)	3.811	1.095	1.000–1.200	0.051
	Triglyceride	0.001 (0.002)	0.572	1.001	0.998–1.004	0.449
	HDL-C	−0.016 (0.012)	1.868	0.984	0.962–1.007	0.172
	Fasting glucose	−0.001 (0.007)	0.012	0.999	0.987–1.012	0.913

To explore the risk factors that are associated with CKD stage 3–5, univariable and multivariable logistics regression models were used. Adjusted variables were chosen on the basis of previous findings and the outcome of the one-way analysis of variance. Age, BMI, triglyceride, HDL-C, fasting glucose, and FIB-4 index group were reported to be associated with the incidence of CKD stage 3–5. A probability value <0.05 was considered to indicate statistical significance. * *p* < 0.05, ** *p* < 0.01. BMI; body mass index, HDL-C; high-density lipoprotein cholesterol.

**Table 3 ijerph-18-06980-t003:** The characteristics of subjects in Study 2 (Baseline).

	Total (*n* = 380)			A Group FIB-4 < 0.80 (*n* = 97)			B Group 0.80 ≤ FIB-4 < 1.02 (*n* = 90)			C Group 1.02 ≤ FIB-4 < 1.32 (*n* = 98)			D Group 1.32 ≤ FIB-4 (*n* = 95)			*p* Value	Tukey HSD	Chi-Square *p* Value
Age (y)	50.5	±	7.0	45.1	±	5.1	49.5	±	6.6	52.5	±	6.1	55.0	±	6.1	<0.001 **	Group A < Group B **, C **, D **	
Gender (female%)	26.6			27.8			26.7			24.5			27.4					0.955
BMI (kg/m^2^)	22.7	±	3.0	23.1	±	2.6	23.2	±	3.2	22.8	±	3.2	21.6	±	2.8	<0.001 **	Group A > Group D	
SBP (mmHg)	123.9	±	15.8	120.9	±	15.2	124.7	±	16.0	124.8	±	16.9	125.2	±	14.7	0.187		
DBP (mmHg)	81.1	±	10.8	80.4	±	11.2	81.6	±	11.2	80.9	±	10.6	81.4	±	10.2	0.88		
Hemoglobin (g/dL)	14.6	±	1.3	14.5	±	1.4	14.7	±	1.1	14.7	±	1.3	14.7	±	1.5	0.811		
Platelet count (×10^4^/μL)	23.1	±	5.1	27.4	±	5.6	24.2	±	3.2	22.2	±	3.1	18.5	±	3.5	<0.001 **	Group A > Group B **, C **, D **	
AST (U/L)	23.3	±	7.8	20.6	±	5.5	21.5	±	5.2	23.0	±	6.3	28.0	±	10.7	<0.001 **	Group A **, B **, C ** < Group D	
ALT (U/L)	25.7	±	14.9	27.2	±	16.7	25.5	±	14.9	23.8	±	12.5	26.1	±	15.3	0.468		
AST/ALT ratio	1.0	±	0.3	0.9	±	0.3	1.0	±	0.3	1.1	±	0.4	1.2	±	0.4	<0.001 **	Group A < Group C, D	
γ-GTP (U/L)	47.1	±	54.9	43.5	±	40.0	40.8	±	47.1	47.7	±	59.6	56.3	±	68.0	0.232		
Total cholesterol (mg/dL)	202.8	±	29.9	200.3	±	26.7	205.0	±	32.7	207.2	±	29.5	198.7	±	30.0	0.162		
Triglyceride (mg/dL)	107.2	±	75.0	106.1	±	70.1	113.3	±	73.2	121.5	±	93.1	87.7	±	54.8	0.013 *	Group C > Group D	
HDL-C (mg/dL)	61.8	±	15.4	59.3	±	13.8	58.5	±	14.4	61.5	±	15.7	67.9	±	16.1	<0.001 **	Group A **, B **, C ** < Group D	
LDL-C (mg/dL)	117.6	±	25.9	118.9	±	24.8	123.3	±	26.4	118.9	±	26.1	109.4	±	24.8	0.002 *	Group A *, B * > Group D	
Fasting glucose(mg/dL)	98.2	±	18.7	95.1	±	10.9	97.0	±	15.2	102.0	±	26.0	98.6	±	18.5	0.063	Group A < Group C *	
HbA1c (%)	5.6	±	0.7	5.5	±	0.3	5.5	±	0.4	5.7	±	0.9	5.6	±	0.9	0.162		
BUN (mg/dL)	13.3	±	3.1	12.8	±	2.9	13.3	±	2.8	13.6	±	2.8	13.6	±	3.9	0.187		
Uric acid (mg/dL)	5.6	±	1.4	5.7	±	1.4	5.6	±	1.3	5.5	±	1.2	5.5	±	1.5	0.486		
Serum creatinine (mg/dL)	0.8	±	0.1	0.8	±	0.1	0.8	±	0.1	0.8	±	0.1	0.8	±	0.1	0.148		
FIB-4 index	1.1	±	0.5	0.7	±	0.1	0.9	±	0.1	1.2	±	0.1	1.7	±	0.5	<0.001 **	Group A < Group B, C, D	
eGFR (mL/min/1.73 m^2^)	77.8	±	10.3	80.8	±	10.1	76.3	±	9.8	76.1	±	10.2	77.9	±	10.4	0.005 *	Group A > Group B, C, D	
Smoking habit (yes/no; *n*, %)	62 (16.3)/318 (83.7)			20 (20.6)/77 (79.4)			10 (11.1)/80 (88.9)			16 (16.3)/82 (83.7)			16 (16.8)/79 (83.2)					0.373
Drinking habit (yes/no; *n*, %)	265 (69.7)/115 (30.3)			68 (70.1)/29 (29.9)			55 (61.1)/35 (38.9)			70 (71.4)/28 (28.6)			72 (75.8)/23 (24.2)					0.175
Antihypertensive drugs (yes/no; *n*, %)	38 (10.0)/342 (90.0)			7 (7.2)/90 (92.8)			11 (12.2)/79 (87.8)			11 (11.2)/87 (88.8)			9 (9.5)/86 (90.5)					0.677
Lipid-lowering drugs (yes/no; *n*, %)	26 (6.8)/354 (93.2)			6 (6.2)/91 (93.8)			8 (8.9)/82 (91.1)			7 (7.1)/91 (92.9)			5 (5.3)/90 (94.7)					0.791
Anti-hyperglycemic drugs (yes/no; *n*, %)	8 (2.1)/380 (97.9)			0 (0.0)/97 (100.0)			1 (1.1)/89 (98.9)			4 (4.1)/94 (95.9)			3 (3.2)/92 (96.8)					0.18

The data are expressed as the mean and standard deviation (SD). We evaluated differences between the four FIB-4 index groups using a one-way analysis of variance. Gender, drinking, and smoking habits; and taking antihypertensive drugs, lipid-lowering drugs, or anti-hyperglycemic drugs were evaluated using the chi-square test. A probability value <0.05 was considered to indicate statistical significance. * *p* < 0.05, ** *p* < 0.01. BMI; body mass index, SBP; systolic blood pressure, DBP; diastolic blood pressure, AST; aspartate aminotransferase, ALT; alanine aminotransferase, γ-GTP; γ-glutamyl transferase, HDL-C; high-density lipoprotein cholesterol, LDL-C; low-density lipoprotein cholesterol, HbA1c; hemoglobin A1c, BUN; blood urea nitrogen, eGFR; estimated glomerular filtration rate.

**Table 4 ijerph-18-06980-t004:** Multivariable analysis of the hazard ratios for CKD stage 3–5 in Study 2.

Model	Characteristics	B (S.E.)	Wald	Hazard Ratio	95% CI	*p* Value
1	FIB-4 index group B	0.440 (0.352)	1.559	1.553	0.778–3.096	0.212
	FIB-4 index group C	0.805 (0.324)	6.187	2.237	1.186–4.219	0.013 *
	FIB-4 index group D	0.340 (0.352)	0.932	1.405	0.705–2.802	0.334
2	FIB-4 index group B	0.128 (0.371)	0.118	1.136	0.549–2.353	0.731
	FIB-4 index group C	0.364 (0.367)	0.984	1.439	0.701–2.954	0.321
	FIB-4 index group D	−0.069 (0.421)	0.027	0.933	0.409–2.131	0.87
	gender	0.278 (0.342)	0.661	1.320	0.676–2.579	0.416
	age	0.054 (0.020)	6.851	1.055	1.014–1.098	0.009 **
	BMI	−0.003 (0.046)	0.005	0.997	0.911–1.090	0.943
	Triglyceride	0.002 (0.001)	3.092	1.002	1.000–1.004	0.079
	HDL-C	−0.010 (0.010)	1.135	0.990	0.971–1.009	0.287
	Fasting glucose	−0.004 (0.006)	0.483	0.996	0.984–1.008	0.487

Differences in the incidence of CKD stage 3–5 among the four FIB-4 index groups were visualized using a Kaplan–Meier curve and log-rank test. A Cox proportional hazards regression models was used to predict the incidence of CKD stage 3–5 for healthy people using the parameters as categorical variables. Hazard ratios were initially adjusted for age, BMI, triglyceride, HDL-C, fasting glucose, and FIB-4 index group at baseline. A probability value <0.05 was considered to indicate statistical significance. * *p* < 0.05, ** *p* < 0.01. BMI; body mass index, HDL-C; high-density lipoprotein cholesterol.

## Data Availability

The contents of the ethics committee’s approval resolution as well as the wording of participants’ written consent do not render open public data access possible. Access to the study’s data set may be requested by contacting the Ethics Committee of Fukuoka University and inform and consent to each participant.

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
