# Peer review of "The Association between Decreased Kidney Function and FIB-4 Index Value, as Indirect Liver Fibrosis Indicator, in Middle-Aged and Older Subjects"

_ijerph, 2021, doi:10.3390/ijerph18136980_

Round 1
Reviewer 1 Report
In this study, Kotoku et al. seek the association between liver fibrosis and kidney function. I reviewed ijerph-1188230 and this revised manuscript does not address any comments I made last time. I will copy&paste my previous comments below as reference. The authors claim that liver fibrosis could be a useful indicator, but individuals in this study do NOT suffer liver fibrosis. No one is suffering kidney dysfunction as well. This is not science to discuss the association between liver fibrosis and kidney function using healthy individuals.
--------------------------------------------
The authors use only healthy individuals and it does not make sense to evaluate fibrosis levels or survival rates. The authors insist statistically significant, but all individuals are healthy anyway. For example, in Table 1, AST is significantly different between groups, but every group is under 30 U/L, which is normal. The authors insist that FIB4 index is significantly different, but generally score <1.45 is recognized as normal. Score >3.25 is risky, but the highest score is 1.74 for D group. It is higher than 1.45 but a lot lower than 3.25, so generally speaking, this is not categorized in liver fibrosis. Since all subjects are healthy, no one is suffering liver fibrosis. Then, what is this study for? Same for eGFR. The authors insist that the difference is significant, but the lowest is 73.3 in group D, which is a little low, but not abnormal. Since group D is highest in age, it makes sense that they have lowest eGFR. Subjects in group D are healthy, so it does not make sense to discuss fibrosis or kidney functions comparing healthy people with healthy people. Survival rates in Figure 3 do not show any difference in all groups because they are all healthy. I cannot understand what the authors want to study and what this manuscript shows.
Reviewer 2 Report
Thank you for resending an updated version of your manuscript.In my opinion, the manuscript notably improved.The current title of the article is definitely more unambiguous and the References section has been prepared in accordance with the Journal guidelines.
The only thing that still raises my doubts and dissatisfaction is the fact that in the chapter "Discussion" there is still no appropriate, short fragment explaining the pathophysiological basis of the existing relationship between NAFLD / NASH and decreased kidney function.
The fragment that was added (lines 243-253) is highlighting the relationship between inflammation, oxidative stress, glucose intolerance, insulin resistance, and the development of glomerulonephritis. The importance of NAFLD in inducing CKD for public health is also emphasized, but still without describing any detailed pathophysiological background of such a relationship. Again, I would suggest for consideration by the Authors adding at least a short paragraph focusing on the strictly proposed pathomechanisms linking both disorders (NAFLD / NASH and renal dysfunction ultimately leading to CKD).
It is my only objection regarding the current version of the paper.
Reviewer 3 Report
Authors analyzed FIB-4 among a group of relatively healthy individuals and found that FIB-4 was associated with CKD prevalence. The work was well performed, and the finding might have its potential value. Comments are as followed.
- To reveal CKD prevalence and incidence in any population, we usually go directly to examine CRTN and eGFR, rather than being guided by FIB-4. Authors may have to re-assess and revise the significance and potential usefulness of FIB-4.
- Age is an absolute risk factor of CKD prevalence. Is FIB-4 still predictive if age factor can be technically effaced by statistical method?
- Readers will be very curious about the finding in Fig.3. In Fig.3, group D performs very well, even better than group B. How to explain this finding?
- Please correct “plate count” in line 130.
Author Response
Please see the attachment.

This manuscript is a resubmission of an earlier submission. The following is a list of the peer review reports and author responses from that submission.
Round 1
Reviewer 1 Report
In this study, Kotoku et al. seek the association between liver fibrosis and kidney function. Although the theme is unique and interesting, procedures are questionable and it is unclear what the authors want to do. The authors use only healthy individuals and it does not make sense to evaluate fibrosis levels or survival rates. The authors insist statistically significant, but all individuals are healthy anyway. For example, in Table 1, AST is significantly different between groups, but every group is under 30 U/L, which is normal. The authors insist that FIB4 index is significantly different, but generally score <1.45 is recognized as normal. Score >3.25 is risky, but the highest score is 1.74 for D group. It is higher than 1.45 but a lot lower than 3.25, so generally speaking, this is not categorized in liver fibrosis. Since all subjects are healthy, no one is suffering liver fibrosis. Then, what is this study for? Same for eGFR. The authors insist that the difference is significant, but the lowest is 73.3 in group D, which is a little low, but not abnormal. Since group D is highest in age, it makes sense that they have lowest eGFR. Subjects in group D are healthy, so it does not make sense to discuss fibrosis or kidney functions comparing healthy people with healthy people. Survival rates in Figure 3 do not show any difference in all groups because they are all healthy. I cannot understand what the authors want to study and what this manuscript shows.
Reviewer 2 Report
Review of the manuscript: ijerph-1188230
The association between decreased kidney function and liver fibrosis in middle-aged and older subjects
The evaluated manuscript is an original article, based on the analysis of the results of cross-sectional and longitudinal studies. The aim of the Authors was to assess whether there is a relationship between liver disorders (FIB4 index value) and kidney function (eGFR) in middle-aged and older patients. Moreover, the influence of other risk factors found in the evaluated patients that may lead to the reduction of eGFR and development of chronic kidney disease (CKD) was analyzed. The demonstration of the phenomenon of the parallel relationship between liver and kidney disturbances is obvious – in pathophysiology, the hepatorenal syndrome has been known for years, and the authors also mention it in the text of their manuscript. However, it would be innovative to use the FIB4 index in the diagnosis of an increased risk of impaired glomerular filtration (eGFR).
It is worth mentioning that statistical analysis described in the manuscript is at a very high level and therefore the obtained results may be considered reliable.
The authors found that liver fibrosis was not an independent risk factor of the prevalence of CKD stage 3-5 in middle-aged and older subjects. FIB4 index and eGFR showed a liner relationship. Moreover, the FIB4 index greater than 1.0 increased the prevalence of CKD stage 3-5 and the authors concluded that the value of the FIB4 index could be a useful indicator to predict for the prevalence of CKD stage 3-5.
My comments and suggestions:
- The title of the work misleads the reader.The authors themselves mention in the text of their manuscript that they did not evaluate liver fibrosis by biopsy and pathomorphological assessment, but they used a biochemical surrogate - the FIB4 index, which is a mathematical formula that combines age, AST and ALT concentrations and the number of platelets. Hence, I suggest to correct the title, eg. “The association between decreased kidney function and FIB4 index value, as indirect liver fibrosis indicator, in middle-aged and older subjects” or to similar ones.
- It is unclear to me why the values of the FIB4 index were adopted differently in the two assessed studies? It would require explanation.
- The fragment describing the pathophysiological rationale for the relationship between chronic hepatitis (NASH / NAFLD-related) and CKD-inducing chronic inflammation (chapter 4.1) is very poor. I would suggest a much broader description of this topic, including pathomechanisms, mediators and cytokines important in the pathogenesis of these chronic inflammatory conditions, with the emphasis on the relationship between the processes taking place in the liver and kidneys . In my opinion, this section is one of the most important in the manuscript, taking into account the revealed correlation between FIB4 and eGFR values. Inn the current version of the manuscript it is too fragmentary.
- The entire "References" chapter has been prepared contrary to the requirements of the IERPH MDPI.There are numerous mistakes - punctation, some journal names are given in full version and they should be in italicized abbreviations, the publication year of citied papers should be written with bold font, and many more.Authors should follow the guidelines:
https://www.mdpi.com/journal/ijerph/instructions#preparation
and
https://www.mdpi.com/authors/references